# Minimally Invasive ALPPS Procedure: A Review of Feasibility and Short-Term Outcomes

**DOI:** 10.3390/cancers15061700

**Published:** 2023-03-10

**Authors:** Luigi Cioffi, Giulio Belli, Francesco Izzo, Corrado Fantini, Alberto D’Agostino, Gianluca Russo, Renato Patrone, Vincenza Granata, Andrea Belli

**Affiliations:** 1Department of General Surgery, Ospedale del Mare, 80147 Naples, Italy; 2Department of General and HPB Surgery, Loreto Nuovo Hospital, 80127 Naples, Italy; 3Division of Epatobiliary Surgical Oncology, Istituto Nazionale Tumori IRCCS Fondazione Pascale-IRCCS di Napoli, 80131 Naples, Italy; 4Department of General Surgery, Pellegrini Hospital, 80134 Naples, Italy; 5Department of General Surgery, San Paolo Hospital, 80125 Naples, Italy; 6Department of General Surgery, University of Campania Luigi Vanvitelli, 80131 Naples, Italy; 7Division of Radiology, Istituto Nazionale Tumori IRCCS Fondazione Pascale-IRCCS di Napoli, 80131 Naples, Italy

**Keywords:** ALPPS, laparoscopic ALPPS, RALPPS

## Abstract

**Simple Summary:**

Associated liver partition with portal vein ligation for staged hepatectomy (ALPPS) represents a recent and promising strategy to perform extensive hepatic resection and limit the risk of post-operative liver failure. Significant morbidity and mortality rates in its pioneering stage has limited acceptance of this treatment. The aim of this review is to evaluate the feasibility, safety, and clinical outcomes of this strategy following application of laparoscopic approach and technical modifications. An evaluation of the data has highlighted that a mini-invasive approach, a less invasive technique in first stage and a better selection of patients could account for potentially better results after ALPPS procedure in terms of blood loss, morbidity, and mortality rate in comparison with outcomes of open series.

**Abstract:**

Background: Associated liver partition with portal vein ligation for staged hepatectomy (ALPPS) represents a recent strategy to improve resectability of extensive hepatic malignancies. Recent surgical advances, such as the application of technical variants and use of a mini-invasive approach (MI-ALPPS), have been proposed to improve clinical outcomes in terms of morbidity and mortality. Methods: A total of 119 MI-ALPPS cases from 6 series were identified and discussed to evaluate the feasibility of the procedure and short-term clinical outcomes. Results: Hepatocellular carcinoma were widely the most common indication for MI-ALPPS. The median estimated blood loss was 260 mL during Stage 1 and 1625 mL in Stage 2. The median length of the procedures was 230 min in Stage 1 and 184 in Stage 2. The median increase ratio of future liver remnant volume was 87.8%. The median major morbidity was 8.14% in Stage 1 and 23.39 in Stage 2. The mortality rate was 0.6%. Conclusions: MI-ALPPS appears to be a feasible and safe procedure, with potentially better short-term outcomes in terms of blood loss, morbidity, and mortality rate if compared with those of open series.

## 1. Introduction

Associated liver partition with portal vein ligation for staged hepatectomy (ALPPS) represents a recent and promising strategy to perform extensive hepatic resection in order to obtain a negative resection margin (R0) and to limit the risk of post-operative liver failure (PHLF).

At present, the standard strategy in the case of extensive hepatic resection with bilobar distribution and presumed insufficient future liver remnant volume (FLRV) is a two-stage hepatectomy with or without preoperative induction of a parenchymal hypertrophy using portal vein embolization (PVE), or, if technically feasible, a one-stage parenchymal sparing hepatectomy. The two-stage hepatectomy has been widely applied with good results in recent decades, allowing an increase of FLRV of 11.9–38% average after 4–8 weeks [1,2]. Nevertheless, patient drop-out, either for insufficient liver hypertrophy or for disease progression between the two stages, still represents the major drawback of such approach. 

Therefore, the surgical community has accepted the introduction of ALPPS as an alternative strategy for R0 resection in a condition of estimated insufficient liver remnant with great expectations. The original ALPPS technique provided two operative times: Stage 1: portal vein ligation without dissection of the remaining structures of the pedicle combined with in situ splitting of the liver;Stage 2: completion of resection of the de-portalized liver via a right or extended right hepatectomy after an accelerated hypertrophy of portalized liver [3].

Nevertheless, after the pioneering phase described in several series [4,5,6], validity and full approval of ALPPS has been strongly debated due to the higher morbidity rate when compared to PVE (33–58 vs. 16%). In fact, poor outcomes in terms of morbidity/mortality data jeopardized potential benefits regarding magnitude and quickness of hypertrophy. 

Over time, in order to preserve the potential benefits of a promising strategy, a more accurate patient selection was pursued and several efforts were made to improve short- and long-term outcomes by promoting different technical variants of “in situ splitting” with the aim of reducing the invasiveness of the first stage:partial ALPPS: limiting the depth and extent of parenchymal transection, allowing a FLR hypertrophy that is comparable to complete transection with a significantly lower morbidity (38.1% vs. 88.9%; *p* = 0.049) and near-zero mortality [7,8] (Figure 1);radiofrequency (RF)- or microwave (MW)-assisted ALPPS (RALPPS or MW-ALPPS) (Figure 2): obtaining a functional liver partition by a “necrotic groove” using RF o MW ablation and allowing a rate of hypertrophy that is comparable to resection associated with a lower morbidity [9,10];tourniquet ALPPS (ALTPS): providing application of a tourniquet around a parenchymal groove of 1 cm in the future transection line [11];mini-ALPPS: combining a partial ALPPS and intraoperative PVE, avoiding dissection of *porta hepatis* [12];hybrid ALPPS: consisting of three steps: in situ splitting, radiological PVE, and completion of hepatectomy [13,14].

Minimally invasive laparoscopic and robotic approaches (MI-ALPPS) have also been advocated in order to assign the well-known benefits of a minimally invasive approach with the aim of reducing the morbidity/mortality rate of this promising surgical strategy. 

The aim of this review is to evaluate the currently available data about the feasibility and safety of MI-ALPPS. Analysis of short-term clinical outcomes of MI-ALPPS in terms of FLVR hypertrophy, length of surgery, blood loss, hospital stay, morbidity, and mortality of the procedure and comparison with the outcomes of open standard ALPPS procedures are the secondary end-points.

## 2. Materials and Methods

### 2.1. Search Strategy

A review of the literature, based on predetermined criteria, was independently performed by two authors (L.C. and A.B.) in 3 databases (PubMed, Scopus, and Cochrane databases) in order to maximize articles capturing data. Boolean search terms ‘ALPPS’ OR ‘Associating liver partition for portal vein ligation for staged hepatectomy’ AND ‘laparoscopic’ OR ‘minimally invasive’ OR ‘robotic’ were used, with no restriction on publishing date. The last search was conducted on October 2022. The identified abstracts were reviewed independently by the two aforementioned authors (L.C. and A.B.) and discrepancies in data collection, synthesis, and analysis were solved by consensus of all authors.

### 2.2. Inclusion Criteria

The inclusion criteria were: (1) English language studies, (2) patients that were operated for an ALPPS procedure with a minimally invasive approach in at least one of the two stages, and (3) studies reporting at least one intra-operative and post-operative outcome as defined below.

### 2.3. Exclusion Criteria

The exclusion criteria included: (1) animal studies; (2) non-English studies; (3) conference abstracts, expert opinions, case reports, editorials, meta-analysis, reviews, and letter to the editors; (4) studies reporting inadequate clinical data; and (5) studies reporting less than 4 patients that were operated on with a minimally invasive approach in at least one of the two stages of an ALPPS procedure. (Figure 3).

### 2.4. Data Extraction and Outcomes

Data extraction was conducted separately by two authors (L.C. and A.B.). Patient characteristics included age, tumor type, and percent ratio sFLVR/weight body (sFLVR/WB%) before Stage 1. Perioperative data included surgical techniques in both stages, length of surgery, estimated blood loss, interval between two stages, and % FLR hypertrophy. Post-operative data included the length of hospital stay, major morbidity rate for two stages, defined as Grade ≥ 3a according to Clavien–Dindo classification [15], and 90-day mortality.

## 3. Results

The outcomes from 119 patients undergoing laparoscopic ALPPS and its variants, extrapolated from six papers that met the inclusion criteria were described in this study. 

The patient’s background features are summarized in Table 1.

Hepatocellular carcinoma (HCC) represented the primary indication for surgery for a total 59 patients (49.6%) undergoing MI-ALPPS, included in three [18,20,21] of the selected studies, although 44 of them were part of a single center study from China [21]. Instead, colorectal liver metastasis (CRLM) represented the most common surgical indication for MI-ALPPS in four of the selected studies [16,17,18,19], for a total of 38 out of 45 patients (84.44%). The intra- and peri-operative data are summarized in Table 2.

In Stage 1, a classic split in situ technique was performed in 10.92% of patients, while modified procedures were performed in 89.07%. The laparoscopic RALPPS technique was performed in 88 patients, robotic RALPPS in 2 patients, laparoscopic classic ALPPS in 13 patients, and laparoscopic partial ALPPS in 16 patients (1 of them robotic and 6 mini-ALPPS, with intraoperative PVE, avoiding hilar dissection). Mini-invasive RALPPS (laparoscopic + robotic) was performed in 90 patients (75.63%) and represented the most common minimally invasive strategy that was applied at the first stage. Partial transection was performed in 16 laparoscopic ALPPS (27%). 

Stage 2 was performed using an open approach in 87 patients, laparoscopic approach in 21 patients, and robotic in in 1 patient. Specifically, 11 right extended hepatectomy and 9 right hepatectomy were completed using laparoscopic approach, and 1 right hepatectomy by robotic approach (Table 3).

A total of nine patients did not complete the second stage of ALPPS because of insufficient liver hypertrophy, progression of disease, or intra-operative complications that occurred during completion of hepatectomy. 

The post-operative data are summarized in Table 4. 

The estimated blood loss during two ALPPS stages was reported for 101 patients from four papers [17,18,19,21]. A total of three of them exclusively investigated a technical variant. In classic ALPPS [21], the range was 110–330 mL and 150–800 mL, in laparoscopic RALPPS [18,21] 20–480 mL and 80–280 mL, and 50–3200 mL and 350–960 mL, respectively, for Stage 1 and 2. The overall median that was observed was 260 mL in Stage 1 and 1625 mL in Stage 2. Details of the perioperative data are summarized in Table 5.

The median length of procedures was 230 min in Stage 1 and 250 in Stage 2.

The median increase ratio of FLVR was +87.8% in MI-ALPPS; a median value of +118% was reported in a series of patients that were undergoing laparoscopic classic ALPPS [20]. 

The time between the two stages ranged from 6 days in a single patient undergoing mini-ALPSS to 36 days in a single patient who underwent a laparoscopic RALPPS. The overall median value was 21 days. 

The hospital stay ranged from 2 to 17 days in Stage 1 and 4 to 17 in Stage 2, with minimum values in Stage 1 for patients that were undergoing the RALPPS technique. The median hospital stay value was 22.5 days.

The overall major morbidity was 8.14% in Stage 1, with 6 cases of biliary fistula requiring percutaneous drainage (Clavien–Dindo Grade IIIa) in the series by Jie et al. [21]. The 90-day mortality was reported in one case after Stage 2 (0.8%) following severe peritonitis after mechanical ischemic obstruction of the small bowel.

## 4. Discussion

PHLF due to insufficient future liver remnant volume (FLRV) represents a key limiting factor for extensive hepatectomy in oncological liver surgery. It is generally agreed that FLRV should be at least 25–30% of the total volume, up to 40% if liver function is compromised because of neoadjuvant chemotherapy or underlying diseases [9]. At the time, the standard strategy to limit PHLF in extensive hepatic resection is a two-stage hepatectomy (with surgical clearance of the FLR in case of bilobar disease) with or without pre-operative induction of parenchymal hypertrophy using PVE or ligation. This approach allows an increase of the estimated FLRV of 11.9–38% after 4–8 weeks. The major morbidity of PVE was seen in 2.2%, while mortality was negligible [1].

Nevertheless, patient drop-out for insufficient liver hypertrophy or for disease progression between the two stages still represents the major drawback of such a strategy. Since its advent, ALPPS appeared as a possible and promising strategy in order to limit both the risk of PHLF in case of extensive liver resection and dropout rate because of progression of disease associated with two-stage hepatectomy.

After its pioneering stage, in which inacceptable morbidity and mortality rates have limited the acceptance of this strategy, a better selection of patients, a refinement in timing of Stage 2, and promotion of several technical variants of the original technique limiting invasiveness of first stage have been proposed in order to take advantage of potential benefits of ALPPS [7,8,9,10,11,12,13,14]. In this direction, minimally invasive approaches, such as laparoscopic and robotic, have been suggested to assign to ALPPS the well-known advantages of laparoscopy, which include reduced blood loss and induction of adhesions between the two stages, minor abdominal wall trauma, faster recovery and shorter hospital stay, and a lower incidence of ascites and liver failure in cirrhotic patients [22,23,24,25]. On the other hand, ALPPS is a technically challenging procedure where a laparoscopic or robotic approach can require both additional expertise in the hands of hepatic surgeon and prolonged operative times. Therefore, the diffusion of minimally invasive ALPPS has been greatly limited because of the related technical difficulties means that only six studies, mainly descriptive case series, including more than four patients that underwent MI-ALPPS, are currently available in the literature.

The analyzed studies showed several discrepancies, dissimilar design, and different levels of accuracy, as MI-ALPPS approach has been associated with different technical variants in Stage 1. Considering the aforementioned limits, some observations can be proposed. 

The current ALPPS strategy has been profoundly subverted compared to the past as it provides, for the lightening of Stage 1, less invasive techniques such as laparoscopy and a reduced or “virtual” liver split in situ to postpone a more aggressive intervention in Stage 2. Indeed, the well-known benefits of laparoscopy could play in synergy with a limited hilar and pericaval dissection, an incomplete liver mobilization, and a reduced parenchymal partition in improving safety of first stage of MI-ALPPS.

Effectively, modified procedures of Stage 1 were performed in 89.07% of patients in the MI-ALPPS series with a prevalence of the mini-invasive RALPPS technique (75.6%) versus 10.16% which was reported for open series in a systematic review by Kawka et al. [26]. Only 23 out of 110 patients (20.9%) that underwent a mini-invasive approach in Stage 1 were submitted to the laparoscopic approach also in Stage 2. These data highlight how the technical difficulties that are connected with Stage 2, which is associated with higher blood loss and a higher incidence of post-operative major complications, currently limit a widespread use of the minimally invasive approach.

Considering the whole MI-ALPPS series, major morbidity rate for Stage 1 was 8.14%, apparently lower than 11% that was reported for open ALPPS series [24], and 23.39% for Stage 2, surprisingly higher than 14.4% that was reported for open series [24]. This higher morbidity rate in MI-ALPPS cases could be related to the influence of series reported by Jie et al. [21] which included HCC as the most frequent indication for surgery (44 patients which 8 Child–Pugh B) taking into account that, on the other side, the majority of the cases in open ALPPS series had been performed for CRLM.

Indeed, a better selection of patients, associated with the lightening of surgical techniques, may have affected the better outcomes in MI-ALPPS cases in four [16,17,18,19] out of the analyzed series in which colorectal liver metastasis (CRLM) represented the main indication for ALPPS procedures, reporting a median major morbidity of 5.96% after Stage 1 and of 15.92% after Stage 2. The mortality rate was 0.6% versus 8.45 for open series. 

This positive trend does appear in accordance with the “paradigm” that was proposed by De Santibanes et al. for which improving short-term outcomes depends mainly on a minor aggressiveness of Stage 1 and full recovery of patients before the second stage, regardless of how the completion surgery was approached [12].

Perihilar and intra-hepatic cholangiocarcinoma (CCA) represented 2% (3/119) of indications in the MI-ALPPS groups. To our knowledge, only three other cases of CCA treatments [27,28,29] have been described for the MI-ALPPS technique, not allowing us to define the role of procedure in the treatment of a tumor where cholestatic features of liver could contribute to the poor outcomes that were reported.

Blood loss reached the lowest value (20 mL) in Stage 1 and the highest value (3200 mL) in Stage 2. These aspects can corroborate that the most challenging surgical step in MI-ALPPS procedures has been postponed in the second stage of the procedure and a full MI-ALPPS in both stages is still rarely performed due to the related technical difficulties.

Interstage time seemed to be shorter and the FLVR ratio higher in open series [16] than in MI-ALPPS series. Nevertheless, 110 out of 119 patients have completed the second stage of ALPPS, highlighting efficiency of a mini-invasive approach in avoiding PHLF, despite the reduced performance in terms of quickness and magnitude of hypertrophy.

Only two studies have reported on the oncologic outcomes for MI-ALPPS [20,21]. In the series of Serenari et al., the median overall survival (OS) did not significantly differ between MI-ALPPS and open-ALPPS (22.6 months versus 17.9 months, *p* = 0.278), while in the study by Jie et al. [21], the median OS of the series was reported to be 22.4 months (3.2–31.4) not allowing us to extrapolate outcomes for different indications.

In addition, the paucity of a homogeneous series in terms of employed surgical techniques and indications for surgery, makes it difficult to assess the specific impact of technical variations of ALPPS on post-operative clinical outcomes. In fact, among the published series focusing on open standard and modified ALPPS, a direct comparison between classic versus partial ALPPS is reported in only two studies. In detail, Chan et al. [30], compared 12 complete versus 13 partial ALPPS for the treatment of HCC, highlighting a more rapid FLR hypertrophy after classic procedures and a non-statistically significant different incidence of post-operative major complications between the two groups in patients with chronic liver disease. On the other hand, Linecker et al. [31] demonstrated that a partial parenchymal partition of at least 50% of the transection line at Stage 1 results in a FLR hypertrophy that is comparable to that reported after complete ALPPS but with a lower rate of minor complications and liver failure (0% vs. 27%; *p* = 0.001) which is similar to those that were reported for MI-ALPPS.

Nevertheless, the lack of comparative data between the open modified ALPPS and modified MI-ALPPS hinder the evaluation of the specific impact of a minimally invasive approach in this setting. In fact, among the studies that focused on MI-ALPPS, merely the early experience of Truant et al. [17] reported a homogeneous series consisting of only five patients undergoing partial ALPPS, which are potentially comparable with the aforementioned open series of Linecker et al. in terms of surgical techniques that were used and indications to surgery. However, the limited sample size that was available in both the modified open and MI-ALPPS precludes any meaningful statistical analysis.

## 5. Conclusions

Over the past few years, a new strategy consisting in a mini-invasive approach, a less invasiveness version of employed techniques in first stage, and a better selection of patients could account for potentially better short-term outcomes after ALPPS procedure in terms of blood loss, morbidity, and mortality rate. Taking into account the actual lack of MI-ALLPS series reported in the literature, the real impact of the minimally invasive approach in the setting of the ALPPS procedures is still to be determined. More comparative data between open ALPPS and MI-ALLPS are needed to determinate the specific impact of the minimally invasive approach.

## Figures and Tables

**Figure 1 cancers-15-01700-f001:**
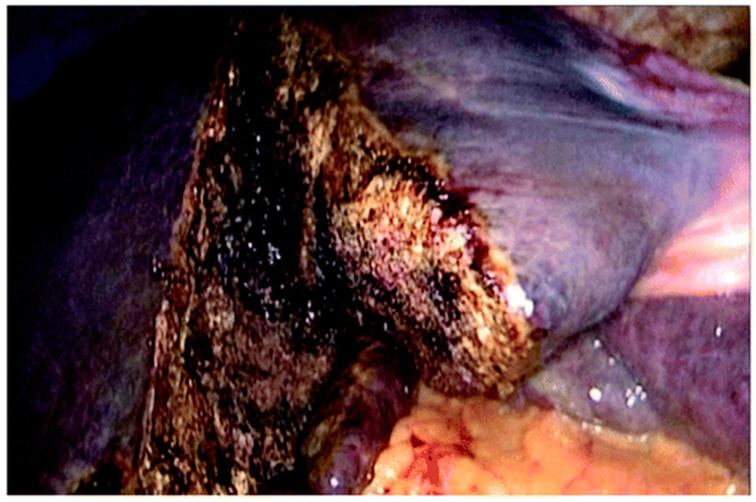
Intraoperative view of Split in situ during laparoscopic partial ALPPS.

**Figure 2 cancers-15-01700-f002:**
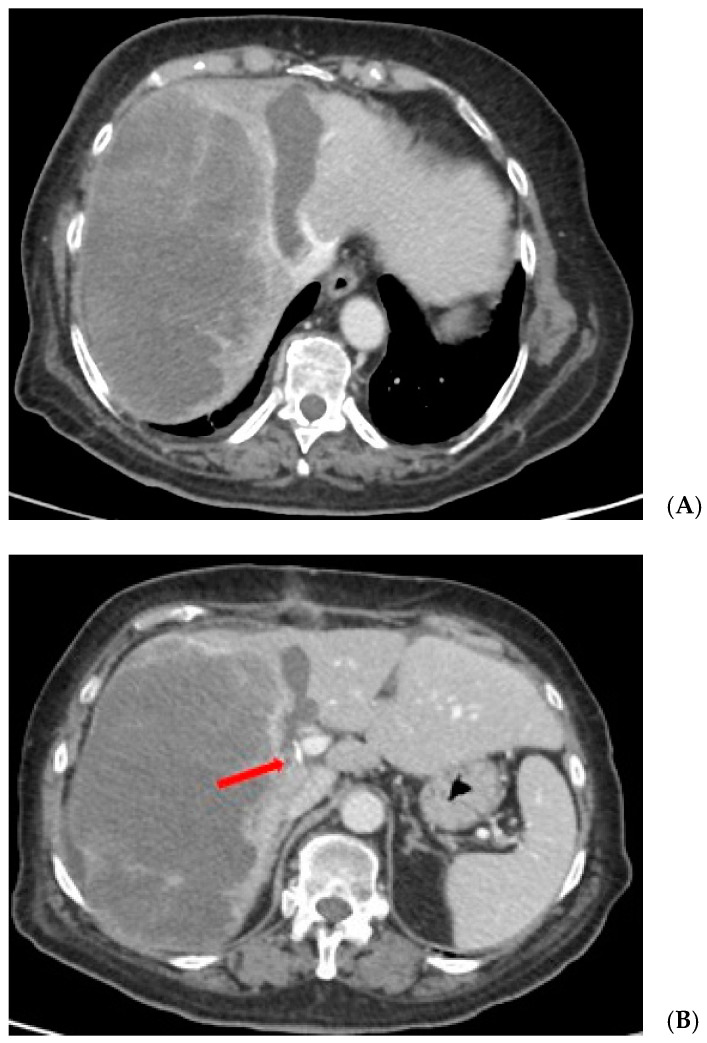
(**A**) CT scan after Stage 1 of RALPPS procedure showing the ablated future transection line. (**B**) CT scan after Stage 1 of RALPPS procedure showing the sectioned right portal vein (red arrow).

**Figure 3 cancers-15-01700-f003:**
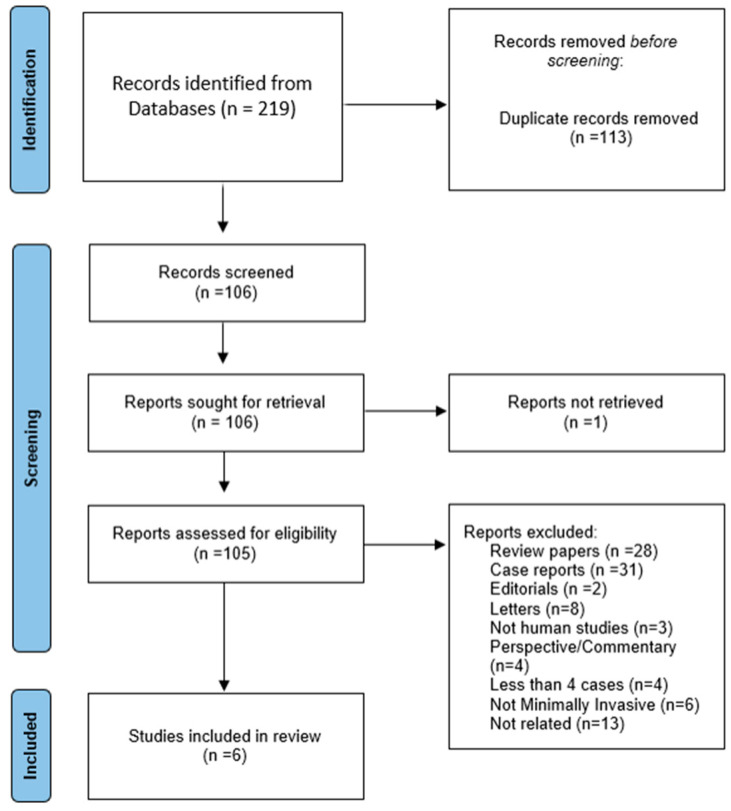
Search strategy and reason for exclusion of studies.

**Table 1 cancers-15-01700-t001:** Patients features.

References	Age (Mean)	*n*	Type of Tumor (*n*)				FLRV/Body Weight Ratio Prior Stage 1
			CRLM	HCC	CCA	other	
Gall et al. [16]	62	5	5	0	0	0	0.5
Truant et al. [17]	60.8	4	4	0	0	0	0.4
Jiao et al. [18]	62.4	26	20	1	0	5	0.23
Machado et al. [19]	58	10	9	0	0	1	0.19
Serenari et al. [20]	64	14	0	14	0	0	0.51
Jie et al. [21]	46.8	60	13	44	3	0	0.78
Overall	59	119	52 (43.5%)	59 (49.5%)	3 (2%)	6 (5%)	0.43

CRLM, Colorectal liver metastasis; HCC, Hepatocellular carcinoma; CCA, Cholangiocarcinoma; FLRV, Future liver remnant volume.

**Table 2 cancers-15-01700-t002:** Perioperative data MI-ALPPS.

References	ALPPS, *n*		Length of Surgery, Median in Min (Range)		Estimated Blood Loss, Median in Mls (Range)		Interstage Time, Median in Days, (Range)	FLVR Hypertrophy, Median in ±% (Range)
	Stage 1	Stage 2	Stage 1	Stage 2	Stage 1	Stage 2		
Gall et al. [16]	Laparoscopic RALPPS *n* = 4	Right Hepatectomy *n* = 4 open	140 (105–140)	-	-	-	23 (12–34)	+62 (53.1–95.4)
Truant et al. [17]	Laparoscopic partial ALPPS *n* = 5	Right extended hepatectomy *n* = 5 open	270 (190–400)	188 (150–280)	250 (100–500)	550 (100–1400)	7.6 (6–13)	+60 (18.6–108.1)
Jiao et al. [18]	Laparoscopic RALPPS *n* = 24 Robotic RALPPS *n* = 2	Right hepatectomy *n* = 19: • 14 open • 4 laparoscopic • 1 robotic Right extended hepatectomy *n* = 5: • 4 open • 1 laparoscopic Not completed *n* = 2	90 (60–125)	180 (110–390)	310 (20–480)	300 (50–3200)	20 (14–36)	+80.7 (67–103.4)
Machado et al. [19]	Laparoscopic ALPPS *n* = 10	Right Hepatectomy *n* = 3 laparoscopic Right extended hepatectomy *n* = 7 laparoscopic	300 (208–340)	180 (140–300)	200 (110–330)	320 (150–800)	21 (9–30)	+118 (42–157)
Serenari et al. [20]	Laparoscopic ALPPS *n* = 7 Laparoscopic mini-ALPPS *n* = 6 Robotic ALPPS *n* = 1 (Laparoscopic partial ALPPS in 11/14)	Right hepatectomy *n* = 2 laparoscopic Right extended hepatectomy *n* = 5 laparoscopic (Converted = 2) Not completed *n* = 7	205 (187–257)	305 (280–360)	-	-	20 (12–27)	+62 (37–91)
Jie et al. [21]	Laparoscopic RALPPS *n* = 60	Right hepatectomy *n* = 32 open Right extended hepatectomy *n* = 28 open	156.8 (102–227)	305.3 (218–407)	165 (80–280)	628 (350–960)	16.4	+45.7
Overall Median (range)	*n* = 119	*n* = 110	230 (60–400)	258.5 (110–407)	260 (20–500)	1625 (50–3200)	21 (6–36)	87.8 (18.6–157)

FLRV, Future liver remnant volume; RALPPS, Radiofrequency-assisted ALPPS.

**Table 3 cancers-15-01700-t003:** Indications and technical features.

	MI-ALPPS
Type of tumors, *n* (%)	
CRLM	52 (43.69%)
HCC	59 (49.57%)
CCA	3 (2.52%)
Other	6 (5.04%)
Split in situ variant Stage 1, *n* (%)	
Classic	13 (10.92%)
Modified	106 (89.07%)
Type of hepatectomy Stage 2, *n* (%)	
Right hepatectomy	60 (54.54%)
Right extended hepatectomy	50 (45.45%)
Left extended hepatectomy	0

CRLM, Colorectal liver metastasis; HCC, Hepatocellular carcinoma; CCA, Cholangiocarcinoma.

**Table 4 cancers-15-01700-t004:** Post-operative data MI-ALPPS.

References	Length of Hospital Stay (Median, Range in Days)			CD Classification Grade ≥ 3a (%)		90 Days Mortality (%)
	Stage 1	Stage 2	Stage 1 + 2	Stage 1	Stage 2	
Gall et al. [16]	-	-		20 - Pulmonary thromboembolism;	-	0
Truant et al. [17]	7 (5–9)	12 (6–18)	19 (11–27)	0	40 - Biliary fistula - Intra-abdominal collection	0
Jiao et al. [18]	9.5 (2–17)	8 (4–32)	27.5 (6–49)	3.85 - Limb compartment syndrome	15.38 - Intra-abdominal collection - Pleural effusion - Post-operative ileus - Small bowel ischemia	3.8
Machado et al. [19]			14 (8–20)	0		0
Serenari et al. [20]	6.5 (4–9)	12 (11–17)	20.5 (15–26)	14.2	8.3	0
Jie et al. [21]	-	23.24	-	13.3	53.3	0
Overall Median (range)	9.5 (2–17)	18 (4–32)	27.5 (6–49)	8.14 (0–20)	23.39 (0–53.3)	0.6 (0–3.8)

CD Classification, Clavien–Dindo Classification.

**Table 5 cancers-15-01700-t005:** Details of perioperative data MI-ALPPS.

	MI-ALPPS
Interstage time, median in days ± IQR	21 (6–36)
FLVR hypertrophy, median in % ± IQR	87.8 (18.6–157)
Length of surgery, Stage 1, median in min ± IQR	230 (60–400)
Length of surgery, Stage 2, median in min ± IQR	250 (110–407)
Estimated blood loss, Stage 1, median in mls ± IQR	260 (20–500)
Estimated blood loss, Stage 2, median in mls ± IQR (Stage 1 and 2)	1625 (50–3200)
CD classification Grade > 3a, Stage 1. median in % ± IQR	8.14 (0–24)
CD classification Grade > 3a, Stage 2, median in % ± IQR	23.39 (0–53.3)
Total length of hospital stay, median in days ± IQR	27.5 (6–49)
90 days mortality, median in % ± IQR	0.8 (0–3.8)

IQR, Inter-quantile ratio; FLRV, Future liver remnant volume; CD Classification, Clavien–Dindo Classification.

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
