# Peer review of "Minimally Invasive ALPPS Procedure: A Review of Feasibility and Short-Term Outcomes"

_cancers, 2023, doi:10.3390/cancers15061700_

Round 1
Reviewer 1 Report
I read the paper of Cioffi al. with great interest. in this systematic review , the authors analyzed the role of mini-invasive surgery associated to ALPPS.
The paper is interesting and well written; the quality of statistical analysis is satisfactory and the relevant literature is adequately covered in the analysis.
Discussion is very complete.
There are only few suggestions to the authors in order to improve the strength of their study:
1. Could You improve the flow diagram of the research with PRISMA layout and respect PRISMA guidelines?
2. In the flow diagram could You define the reason for not relevance of the articles (eg. animal study, letter to editor...)
3. I suggest to cite in the "hybrid ALPPS" paragraph the paper "Hybrid Partial ALPPS: a Feasible Approach in Case of Right Trisegmentectomy and Macrovascular Invasion."
4. I suggest to add to the analysis more informations about patient survival and disease free survival
Thanks
Author Response
The authors thank the Reviewer for the comments and the important suggestions. We modified the manuscript as suggested. Point by point answer are detailed below:
- Could You improve the flow diagram of the research with PRISMA layout and respect PRISMA guidelines?
The authors thank the Reviewer for the observation. We decided to eliminate a comparative analysis and to trasform the paper in observational review because of the still limited data available in the literature on MI-ALLPS. We focused the manuscript on the technical aspects and the short-term outcomes of MI-ALPPS. We improved the flow diagram following PRISMA guidelines.
- In the flow diagram could You define the reason for not relevance of the articles (eg. animal study, letter to editor...)
We detailed the reasons for exclusion as requested.
- I suggest to cite in the "hybrid ALPPS" paragraph the paper "Hybrid Partial ALPPS: a Feasible Approach in Case of Right Trisegmentectomy and Macrovascular Invasion."
The authors thank the Reviewer for the observation. We added the suggested reference.
- I suggest to add to the analysis more informations about patient survival and disease freesurvival
The authors thank the Reviewer for the observation. We added additional data on the oncological outcomes of MI-ALPPS as requested, extrapolating them from two paper where they are available.
Reviewer 2 Report
Cancers_2134876
The authors reviewed the feasibility of minimally invasive-laparoscopic and robotic-approach (MIS-ALPPS), and compared MIS-ALPPS (n=59) with open-ALPPS (n=1088) in only short-term outcomes.
They well summarized and reviewed the current history of ALPPS and mentioned the feasibility of MIS-ALPPS. MIS-ALPPS would seem to be better than reduced morbidity and mortality rates. However, there are still many concerns about this new type of operation, such as the indication of the MIS approach and oncological long-term outcomes.
Major comments
1. Although MIS-ALPPS would seem to be better than reduced morbidity and mortality rates, reduced morbidity and mortality rates is owed to not only MIS ALPPS but also modified approach such as partial ALPPS, radiofrequency ALPPS, tourniquet ALPPS, mini-ALPPS, and hybrid ALPPS. The authors should compare modified MIS-ALPPS with modified open ALPPS. Modified MIS-ALPPS was performed in about 46 patients (77.6%) and modified open ALPPS was performed in 115 patients (10.16%). That also resolves the sample size problem between MIS-ALPPS and open ALPPS (n=59 vs. n=1088)
2. In Table.5, p values were not described. Only significant or not significant was described. Student t test was used for statistical analysis, did the authors confirm these data were parametric data?
3. The data of open ALPPS was referred from only one review paper. It is not scientific.
4. The authors should describe all the reported open ALPPS variants and references. The referred paper of open ALPPS published by Kawka et al. is very similar to the present review. This review added only 1 paper data of Serenari et al. published in 2020.
5. The review written by Kawka et al. excludes sample size of less than 20 patients in the open ALPPS group, regardless of including papers of less than 20 patients in the MIS-ALPPS group.
6. This review was not recognized as a systematic review correctly, because meta-analysis was not performed.
Minor comments
The order seems to be reversed, references 15 and 16.
Author Response
REVIEWER 2:
The authors thank the Reviewer for the comments and the important suggestions. We modified the manuscript as suggested. Point by point answer are detailed below:
Major comments
- Although MIS-ALPPS would seem to be better than reduced morbidity and mortality rates, reduced morbidity and mortality rates is owed to not only MIS ALPPS but also modified approach such as partial ALPPS, radiofrequency ALPPS, tourniquet ALPPS, mini-ALPPS, and hybrid ALPPS. The authors should compare modified MIS-ALPPS with modified open ALPPS. Modified MIS-ALPPS was performed in about 46 patients (77.6%) and modified open ALPPS was performed in 115 patients (10.16%). That also resolves the sample size problem between MIS-ALPPS and open ALPPS (n=59 vs. n=1088)
The authors thank the Reviewer for the observation. This is a very relevant point. We do agree that reducing the extent of parenchymal transection from the original ALPPS technique by the adoption of modified approaches such as mini-ALPPS, RALPPS etc. can have a direct impact on the morbidity rate of the first stage. Similarly, the adoption of a minimally invasive approach at the first stage may have played a role by adding the well-known benefit of laparoscopy. Therefore, it is hard to discriminate which one of the players (MIS approach or modified surgical technique) had the prominent role in reducing the postoperative morbidity with the current available evidences.
Therefore. as suggested also in point 3, we decided to eliminate the comparative analysis with the results of the review written by Kawka et al. because of the still limited data available in the literature on MI-ALLPS. We focused the manuscript on the technical aspects and the short-term outcomes of MI-ALPPS.
- In Table.5, p values were not described. Only significant or not significant was described. Student t test was used for statistical analysis, did the authors confirm these data were parametric data?
The authors thank the Reviewer for the observation. See point 1
- The data of open ALPPS was referred from only one review paper. It is not scientific.
The authors thank the Reviewer for the observation. See point 1
- The authors should describe all the reported open ALPPS variants and references. The referred paper of open ALPPS published by Kawka et al. is very similar to the present review. This review added only 1 paper data of Serenari et al. published in 2020.
The authors thank the Reviewer for the observation. As stated in the previous answer we decided to focus our manuscript on the technical aspects and the short-term outcomes of MI-ALPPS. We also added the data from an additional paper from Ji et al. which included 60 additional patients treated by MI-RALPPS at the first stage.
- The review written by Kawka et al. excludes sample size of less than 20 patients in the open ALPPS group, regardless of including papers of less than 20 patients in the MIS-ALPPS group.
The authors thank the Reviewer for the observation. We included studies reporting more than 4 patients -and not more than 20 patients- since MI-ALPPS is a relatively new surgical option and only 6 papers including at least 4 patients reported on this innovative technical option.
- This review was not recognized as a systematic review correctly, because meta-analysis was not performed.
The authors thank the Reviewer for the observation. We agree and we have turned the paper into an observational paper
Minor comments
The order seems to be reversed, references 15 and 16.
The authors thank the Reviewer for the observation. We modified the references in the text.
Round 2
Reviewer 2 Report
We have raised the question that the authors should compare modified MIS-ALPPS with modified open ALPPS. Modified MIS-ALPPS was performed in about 46 patients (77.6%) and modified open ALPPS was performed in 115 patients (10.16%). However, the fundamental descrpition is not changed.
The amounts of blood loss during 1st stage and 2nd stage have a big difference. This suggest that MI ALPPS is limited in 1st stage, but it is difficutl to apply it in the 2nd stage.
Round 3
Reviewer 2 Report
This paper was well written